# BIDIRECTIONAL TEMPORAL DIFFUSION MODEL FOR TEMPORALLY CONSISTENT HUMAN ANIMATION

**Tserendorj Adiya**[3,*]   **Jae Shin Yoon**[2]   **Jungeun Lee**[1]   **Sanghun Kim**[1]   **Hwasup Lim**[1]

[1]Korea Institute of Science and Technology   [2]Adobe   [3]AI Center, CJ Corporation

ts.adiya@cj.net, jaeyoon@adobe.com, {092599,kei97103,hslim}@kist.re.kr

## ABSTRACT

We introduce a method to generate temporally coherent human animation from a single image, a video, or a random noise. This problem has been formulated as modeling of an auto-regressive generation, *i.e.*, to regress past frames to decode future frames. However, such unidirectional generation is highly prone to motion drifting over time, generating unrealistic human animation with significant artifacts such as appearance distortion. We claim that *bidirectional* temporal modeling enforces temporal coherence on a generative network by largely suppressing the appearance ambiguity. To prove our claim, we design a novel human animation framework using a denoising diffusion model: a neural network learns to generate the image of a person by denoising temporal Gaussian noises whose intermediate results are cross-conditioned bidirectionally between consecutive frames. In the experiments, our method demonstrates strong performance compared to existing unidirectional approaches with realistic temporal coherence.

## 1 INTRODUCTION

Humans express their own space-time continuum in the form of appearance and motion. While existing generative models Isola et al. (2017); Sarkar et al. (2021) have been successful to restore the space, *i.e.*, high-quality image generation with diverse human appearance, they often fail to decode the time, *e.g.*, temporally incoherent human motion. In this paper, we introduce a method for temporal modeling of a generative network to synthesize temporally consistent human animations. Our method can generate a human animation from three different modalities: a random noise, a single image, and a single video as shown in Figure. 1. Such generated human animations enable a number of applications including novel content creation for non-expert media artists and pre-visualization of human animation that can be further refined by professional video creators.

The temporal modeling for human animation has been often formulated as a video auto-regression problem: using past frames as a condition to decode future frames. While such unidirectional generation (forward auto-regression) has shown smooth animation results, it often suffers from texture drifting, *e.g.*, the texture on the clothing of a person such as a skirt in Figure 2 is largely distorted along its dynamic movements. This is mainly due to the significant motion-appearance ambiguity where there exist infinite solutions to decide the future state of human appearance even with the same motion, which amplifies the artifacts (e.g., distortion) over time.

To suppress such motion-appearance ambiguity, we model a human appearance bidirectionally: a generative network decodes the human appearance in the context of both forward and backward image regression whose intermediate features are cross-conditioned over time. Our key observation is that the bidirectional temporal consistency in feature space highly suppresses the motion-appearance ambiguity, which prevents from the texture drifting while maintaining its temporal smoothness.

We realize the idea of bidirectional temporal modeling by utilizing a generative denoising diffusion model Ho et al. (2020). A denoising network learns to iteratively remove temporal Gaussian noises to generate the human animation guided by conditioning poses and appearance style. Inspired by message passing algorithms in dynamic programming Felzenszwalb & Zabih (2010); Arora et al.

---

*Work done during internship at Korea Institute of Science and Technology.

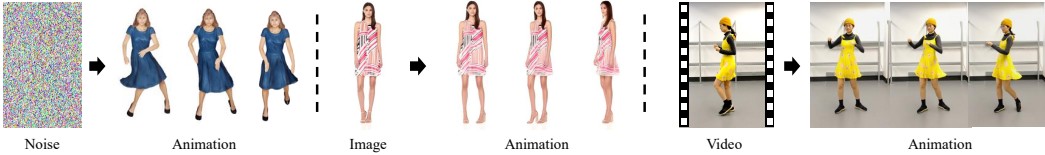

Figure 1: Our method generates temporally coherent human animation from various modalities.

(2009), we recursively cross-condition the intermediate results between consecutive frames in a bidirectional way as shown in Figure 3; where the temporal context of human appearance is locally consistent for consecutive frames at the first denoising step, and it is progressively refined at every denoising iteration to be globally coherent for entire frames.

In the experiments, we demonstrate that our bidirectional denoising diffusion model generates human animations from a single image with a strong temporal coherence, outperforming the results from unidirectional generative models. We also show that learning from multiple frames, *i.e.*, a person-specific video, can further improve the physical plausibility of the generated human animation. Finally, we showcase that our method can generate human animations with diverse clothing styles and identities without any conditioning images.

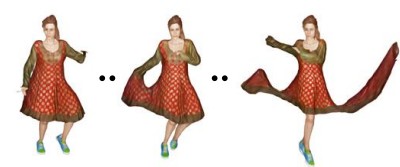

Figure 2: Results from a unidirectional generative model with texture drifting over time.

**Contribution** (1) We propose a bidirectional temporal diffusion model that can generate temporally coherent human animation from random noise, a single image, or a video. (2) Inspired by dynamic message passing algorithms, we introduce the feature cross-conditioning between consecutive frames with recursive sampling, which allows embedding the motion context on the iterative denoising process in a locally and globally consistent way. (3) We quantitatively and qualitatively demonstrate that our method shows a strong temporal coherence compared to existing unidirectional methods. For an accurate evaluation, we newly create high-quality synthetic data of people in dynamic movements using graphics simulation, which provides ground-truth data, *i.e.,* different people in the perfectly same motion.

## 2 RELATED WORKS

**Human Motion Transfer** Given a sequence of guiding body poses and the style of human appearance, it aims to generate the human animation that satisfies the conditioning motion and style. Many existing pose transfer methods have utilized 2D keypoints as conditioning body pose maps Chan et al. (2019); Balakrishnan et al. (2018); Esser et al. (2018); Liu et al. (2019a). However, these approaches often fail to extract the physical implications from the keypoints maps, resulting in temporally unnatural human animation.

To address this motion consistency issue, methods such as EDN Chan et al. (2019), V2V Wang et al. (2018), and DIW Wang et al. (2021) leveraged Markovian independence to generate auto-regressive frames. These approaches utilize Densepose Güler et al. (2018) as a 2D pose conditioning and learn motion-dependent appearance for a specific person, producing realistic animation results for unseen motions. Recent advancements in this area involve embedding 3D velocities from the SMPL Loper et al. (2015) model as pose conditioning Yoon et al. (2022), leading to the better generation of complex transformations. However, these methods require extensive training on the videos of a single individual, limiting their generalizability to diverse people.

To synthesize human animations of diverse people using a single model, several works have studied human motion transfer from a single image. Solutions include applying affine transformations Balakrishnan et al. (2018); Zhou et al. (2019), flow-based warping Wang et al. (2019); Zhao & Zhang (2022); Siarohin et al. (2021; 2019), or assuming a base 3D human model and texture mapping with DensePose Neverova et al. (2018); Huang et al. (2021) or the SMPL model Li et al. (2019); Liu et al. (2019b). However, these methods struggle to represent diverse surface transformations in clothing, *i.e.,* the clothing texture looks static even under the pose changes, resulting in unnatural animations.

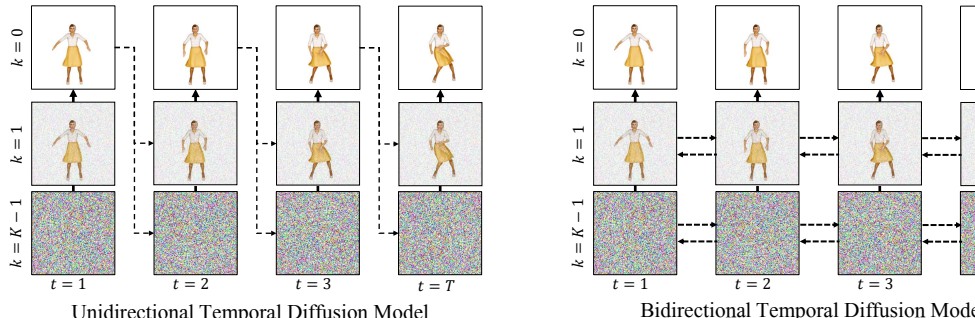

Figure 3: The left illustration represents a unidirectional diffusion model, and the right one provides an overview of our proposed bidirectional temporal diffusion model (BTDM). The dotted arrows indicate the direction of conditioning, and $k$ and $t$ represent the denoising step and time interval, respectively.

**Generative Diffusion Models** Recently, diffusion models have demonstrated outstanding performance in high-quality image generation Ho et al. (2020); Nichol & Dhariwal (2021); Song et al. (2020b), text-to-image translation Rombach et al. (2022); Preechakul et al. (2022); Saharia et al. (2022a); Ramesh et al. (2021), image super-resolution Saharia et al. (2022b), image restoration Kawar et al. (2022); Wang et al. (2022) view synthesis Watson et al. (2022). Compared to generative adversarial networks (GANs) Isola et al. (2017); Karras et al. (2019), diffusion models enable more stable training and reduced mode collapse, leading to diverse and high-quality generation results.

The initial diffusion model was based on Song's Song & Ermon (2019) score-matching approach, which estimates gradients using Langevin dynamics to infer data distributions. Subsequently, the DDPM Ho et al. (2020) method was introduced, leveraging weighted variational bounds and becoming widely adopted. Later, NCSN Song & Ermon (2019) and its equivalent from ODE Song et al. (2020b) emerged, presenting a more general form.

One notable drawback of Markov Chain Monte Carlo (MCMC) based inference in diffusion models is the longer inference time compared to GANs. DDIM Song et al. (2020a) addresses this issue by interpreting the diffusion process as an implicit function that significantly reducing sampling time while preserving generation quality.

Diffusion models, as referenced in recent studies such as Ho et al. (2022b); Yang et al. (2023); Ho et al. (2022a); Singer et al. (2022); Zhou et al. (2022); Esser et al. (2023); Guo et al. (2023), have become increasingly popular in the field of video generation. These models often utilize techniques like cross-attention and 3D U-Nets to ensure videos remain consistent over time. Despite their potential, most of these methods face challenges in generating longer videos without encountering issues like shape drifting and appearance jitters. This paper presents a new and practical approach to overcome these limitations. Our method, which is distinct from previous work, employs a 2D-Unet based framework to create temporally coherent animations. Notably, our approach is effective in generating animations of humans and is not constrained by video length.

## 3  METHOD

Conventional Denoising Diffusion Probabilistic Models (DDPM) work by gradually diffusing isotropic Gaussian noise onto a data sample $y \in \mathcal{D}$ across $K$ steps along a Markovian chain. The process is reversed, such that $y$ is approximated from the $\mathcal{N}(0, I)$ distribution.

One can extend these conventional DDPM to generate a human animation driven by a sequence of human pose maps $\mathcal{S} = \{s_1, ..., s_T\}$ (*e.g.*, densepose Güler et al. (2018)) in an auto-regressive way. For example, a network is designed to generate future frames dependent on previous frames by gradually diffusing isotropic Gaussian noise onto the training sample $y_t \in Y = \{y_1, ..., y_T\}$ under the conditional Markovian independence, *i.e.*, $p(Y) = \prod_{t=1}^{T} p(y_t|y_{t-1}; s_t \in \mathcal{S})$. However, such autoregressive models often suffer from texture drifting due to the motion-appearance ambiguity that is inherent in unidirectional prediction.

To suppress the motion-appearance ambiguity, we design a bidirectional temporal diffusion model (BTDM) as shown in Figure 3. BTDM learns motion-dependent appearances in both forward and backward directions along the time axis. The denoising results from each step in either time direction serve as mutual conditions for generating human animation. Our model can generate realistic animations unconditionally, as well as conditionally from a single image or video.

## 3.1 BIDIRECTIONAL TEMPORAL DIFFUSION MODEL

Given a pose sequence $S$ and its corresponding image sequence $Y$, modeling their mapping bidirectionally along the time axis that follows Markovian independence results in:

$$p_f(Y|S) := \prod_{t=1}^{T} p(y_t|y_{t-1}, s_t), \qquad p_b(Y|S) := \prod_{t=1}^{T} p(y_{t-1}|y_t, s_{t-1}) \tag{1}$$

In this setup, $p_f$ represents the forward direction along the time axis, and $p_b$ signifies the backward direction. We define a marginal distribution with isotropic Gaussian process that gradually adds increasing amounts of noise to the data sample as the signal-to-noise-ratio $\lambda(\cdot)$ decreases, following Salimans & Ho (2021):

$$q(y_t^{1:K}|y_t^0) := \prod_{k=1}^{K} q(y_t^k|y_t^{k-1}), \qquad q(y_t^k|y_t^{k-1}) := \mathcal{N}(y_t^k; \sqrt{\sigma(\lambda(k))}y_t^{k-1}, \sigma(-\lambda(k))\mathbf{I}) \tag{2}$$

where $\sigma(\cdot)$ is the sigmoid function, $K$ is the number of diffusion step, and $\mathbf{I}$ denotes the identity.

Both the motion-dependent appearance distribution in Equation 1 and the diffusion process in Equation 2 follow a Markovian chain. Ideally, we should predict $y_t$ and $y_{t-1}$ using perfectly denoised $y_{t-1}^0$ (in the forward direction) or $y_t^0$ (in the backward direction) as conditions. However, such perfectly denoised images are not available during inference, which leads to overfitting to the training data and amplifies the error from motion-appearance ambiguity. For this reason, we integrate these two independent Markovian chains as follows:

$$p_f(Y^k|S) := \prod_{t=1}^{T} p(y_t^k|y_{t-1}^k, s_t), \qquad p_b(Y^k|S) := \prod_{t=1}^{T} p(y_{t-1}^k|y_t^k, s_{t-1}), \tag{3}$$

where by utilizing the noisy $y_k$ as a condition, we concurrently diminish the reliance of motion-dependent appearance generation on the preceding frame and avert overfitting to the condition, thereby alleviating artifacts when generating unseen conditions and improving the model's generalization performance. This approach also yields more temporally consistent animations by highly limiting the motion diversity between consecutive frames.

Although $p_f$ and $p_b$ are independent, $p(y_t|y_{t-1})$ and $p(y_{t-1}|y_t)$ are concurrently defined on the time axis $t$. This allows us to optimize both probabilities simultaneously. Therefore, the objective function for training is defined as follows:

$$
\begin{aligned}
L = \mathbb{E}_{t \sim [1,T], k \sim [1,K], y^k \sim q_k, d_f, d_b, c} \Big[ \frac{1}{2} \big( &||f_\theta(y_t^k, y_{t-1}^k, \lambda(k), s_t, c, d_f) - y_t^0||_2^2 \\
&+ ||f_\theta(y_{t-1}^k, y_t^k, \lambda(k), s_{t-1}, c, d_b) - y_{t-1}^0||_2^2 \big) \Big]
\end{aligned}
\tag{4}
$$

where $f_\theta$ is a neural network whose task is to denoise the frame $y_{t-1}^k$, $y_t^k$ given a different noisy frame $y_t^k$, $y_{t-1}^k$ and given pose $s_t$, $s_{t-1}$. The $\lambda$ is the log signal-to-noise-ratio function dependent on $k$, and $c$ is a single image condition that determines the appearance of a target person. The notation $d_f, d_b$ are learnable positional encoding vectors for distinguishing temporal direction. Following the method used in Ramesh et al. (2022), we adapt our model to predict clean images instead of noise.

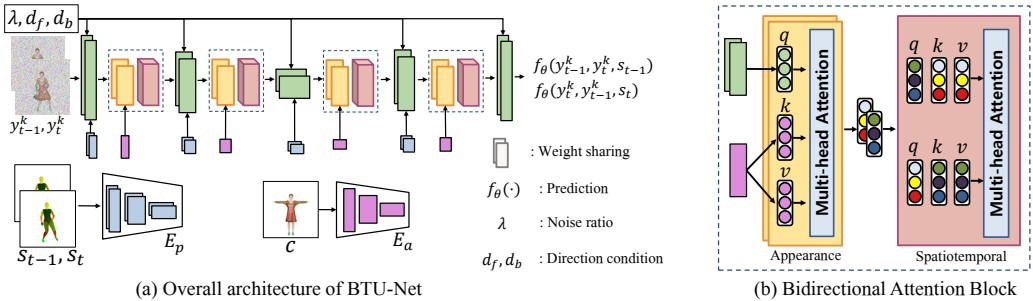

(a) Overall architecture of BTU-Net

(b) Bidirectional Attention Block

Figure 4: The illustration of (a) our BTU-Net and (b) bidirectional attention block. The dotted squares in (a) represents bidirectional attention block. The small blue and pink squares in (a) indicate the intermediate feature of $E_p$ and $E_a$, respectively.

## 3.2 BIDIRECTIONAL TEMPORAL U-NET

To enable BTDM, we construct a Bidirectional Temporal U-Net (BTU-Net) by modifying the U-Net architecture, as shown in Figure 4. This architecture consists of a network, $E_a$, that encodes a single image condition $c$; another network, $E_p$, that encodes poses $s$ corresponding to $t-1$ and $t$; and a pair of U-Nets, $f_\theta$, that accept $y_{t-1}^k$ and $y_t^k$ as input and predict the denoised human images temporally in both forward and backward directions. The multi-scale intermediate features are modulated by pose features, noise ratio $\lambda$, and temporal direction vectors ($d_f$ and $d_b$) using existing Feature-wise Linear Modulation layer (FiLM) Perez et al. (2018). This pair of U-Nets shares weights and applies attention between the features encoded by $E_a$ and intermediate features of $f_\theta$, as shown in the bidirectional attention block in Figure 4(b), which is composed of appearance and spatiotemporal block. Each block utilizes multi head attention Vaswani et al. (2017).

**Appearance Block** The yellow box in Figure 4(b), the appearance attention block is composed of parallel multi-head attention mechanisms with weight sharing. For the multi-head attention, features from times $t$ and $t-1$ of $f_\theta$ are used as the query, while the appearance features of $E_a$ are employed as the key and value. This block is specifically designed to learn the correlation between the appearance features encoded by $E_a$ and the intermediate features of $f_\theta$. Through the multi-head attention mechanism, it attends to global context correlations, facilitating the effective transfer of appearance features to $f_\theta$ even during rapid actions (those with substantial pose changes relative to frame intervals). This, in turn, enables the generation of novel poses effectively.

**Spatiotemporal Block** The spatiotemporal attention block takes the output feature pairs from the appearance attention block as inputs and, as illustrated in Figure 4 (b), performs cross attention. Such a structure effectively enables the learning of the temporal correlations of spatial features between times $t$ and $t-1$. Furthermore, by making the features for generating $t$ dependent on $t-1$ and vice versa, a temporally bidirectional structure is achieved. This design facilitates the efficient learning of temporal correlations.

We adopt the bidirectional attention block for the feature at specific resolutions, *i.e.*, $32 \times 32$, $16 \times 16$, and $8 \times 8$. More details on the architecture can be found in Appendix B.3.

## 3.3 TRAINING AND INFERENCE FOR VARIOUS TASKS

**Single Image Animation** Our BTDM, trained on multiple videos, can be directly applied to generate realistic human animation results for unseen people and poses. Similar to existing one-shot generation methods Wang et al. (2018), we further fine-tune our BTU-Net on the given single image to enhance the visual quality. For this, the conditioning image $c$ is set as a single image sequence $Y = \{c\}$ and the pose sequence $S = \{g(c)\}$, where $g(\cdot)$ is a pose estimation function (*e.g.* Dense-Pose). This setup aligns with the training process outlined in Equation 4.

**Person-Specific Animation** Our method can be applied to the task of generating novel animations by training a single person's video. To adapt our method to this task, we train our BTDM framework using the objective function from Equation 4, excluding the image condition $c$.

**Unconditional animation** Moreover, our method facilitates the creation of temporally consistent animations without any appearance-related conditions. For such unconditional generation, we trained our model with the condition $c$ set to $\emptyset$ of Equation 4.

During the inference stage, to effectively utilize our BTDM, we employ a bidirectional recursive sampling method across all tasks. More details about this method can be found in Appendix A.

## 4 EXPERIMENTS

We validate our bidirectional temporal diffusion model on two tasks: generating human animation from a single image and generating human animation by learning from a person-specific video. We also show that our model can generate diverse human animation with an unconditional setting (i.e., generating human animation from random noise).

### 4.1 SINGLE IMAGE ANIMATION

**Dataset** We use two datasets that can effectively validate the quantity and quality of temporal coherence in the generated human animation. 1) Graphics simulation: for quantitative evaluation, we construct a high-quality synthetic dataset using a graphics simulation tool for soft 3D clothing animation Reallusion (b) which provides perfect ground truth data for the motion transfer task (*i.e.,* different people in the exact same motion, which does not exist from real-world videos) with physically plausible dynamic clothing movements. The dataset includes a total of 80 training videos and 19 testing videos, each of which lasts 32 seconds at 30 FPS. We customize the 3D human appearance using CharacterCreator Reallusion (a), and we use Mixamo motion data Adobe for animation. The pose map is obtained by rendering the IUV surface coordinates of a 3D body model (*i.e.,* SMPLLoper et al. (2015)). Please see the appendix D for more details of our graphics simulation data. 2) UBC Fashion dataset Zablotskaia et al. (2019): it consists of 500 training and 100 testing videos of individuals wearing various outfits and rotating 360 degrees. Each video lasts approximately 12 seconds at 30 FPS. We apply DensePose Güler et al. (2018) to obtain pose UV maps. We use this dataset for the qualitative demonstration on real images since it does not provide ground truth data with the exact same motion.

**Baselines** We compare our method to existing unidirectional temporal models: Thin-Plate Spline Motion Model for Image Animation (TPSMM) Zhao & Zhang (2022) and Motion Representations for Articulated Animation (MRAA) Siarohin et al. (2021) are designed to predict forward optical flow to transport the pixel from a source to target pose, following a rendering network. Both methods were trained on each dataset from scratch using the provided scripts and recommended training setup. All methods are trained at a resolution of 256×256.

**Metric** To evaluate the quality of the generated human animations, we employ five key metrics: 1) SSIM (Structural Similarity Index) Wang et al. (2004): quantifies the structural similarity between the generated and ground truth images based on local patterns of pixel intensities and contrast spaces. 2) LPIPS (Learned Perceptual Image Patch Similarity) Zhang et al. (2018): cognitive similarity between synthesized images and ground truth images by comparing the perceptual features extracted from both, utilizing a pre-trained deep neural network. 3) tLPIPS (Temporal Learned Perceptual Image Patch Similarity) Chu et al. (2020): extends the LPIPS measure to temporal domain, evaluating the plausibility of change across consecutive frames. It is defined as $\text{tLPIPS} = ||\text{LPIPS}(y_t, y_{t-1}) - \text{LPIPS}(g_t, g_{t-1})||$, where $y$ and $g$ represent the synthesized and ground truth images, respectively. 4) tOF Chu et al. (2020): pixel-wise difference of the estimated optical flow between each sequence and the ground truth. 5) FID (Fréchet Inception Distance) Heusel et al. (2017): measures the distance between the distributions of synthesized and real images in the feature space of a pre-trained Inception network.

**Result** The quantitative results for the graphics simulation data are presented in Table 1. Our BTDM method outperforms other methods in all metrics. As can be seen in Figure 5, our method closely resembles the source image, and the appearance changes depending on the movement are more realistic than other baseline methods. TPSMM and MRAA undergo significant artifacts such as texture distortion and blur due to errors in the forward optical flow prediction. In particular, the models from baseline methods highly confuse on the motion with large dynamics. The same trend is observed in the UBC Fashion data. Specifically, when the appearance of a driven video significantly

Table 1: Quantitative results for single image animation tested on simulation data.

| Methods | SSIM↑ | LPIPS↓ | tLPIPS↓ | tOF | FID↓ |
|---------|-------|--------|---------|-----|------|
| MRAA Siarohin et al. (2021) | 0.894 | 0.140 | 0.011 | 13.58 | 67.68 |
| TPS Zhao & Zhang (2022) | 0.915 | 0.077 | 0.005 | 11.92 | 48.76 |
| Ours | **0.958** | **0.036** | **0.003** | **8.93** | **11.14** |

Figure 5: Qualitative comparisons for the single image animation task on graphics simulation (left) and UBC Fashion data (right).

differs from the source image in TPSMM or MRAA methods, abnormal artifacts often occur such as the loss of identity. Moreover, our method is found to preserve fine details considerably better.

## 4.2 PERSON-SPECIFIC ANIMATION

**Dataset** To evaluate the performance of our method in the task of person-specific animation, we use five videos from Yoon et al. (2022). Each video comprises between 6K and 15K frames, featuring a person performing a diverse range of dynamic actions. The pose UV map is obtained using DensePoseGüler et al. (2018).

**Baseline** We compare our method to V2V Wang et al. (2018), EDN Chan et al. (2019), HFMT Kappel et al. (2021), DIW Wang et al. (2021), and MDMT Yoon et al. (2022), which utilized a generative network in a temporally unidirectional way. All methods were trained on the training set of each video and evaluated on the test set.

**Metrics** We use SSIM, LPIPS, and tLPIPS as used in Section 4.1.

**Result** The evaluation results for our method and the baselines on the test sequences from the five videos are displayed in Table 2. Our approach exhibits a performance that is either comparable to or surpasses that of other state-of-the-art methods in the LPIPS and tLPIPS metrics. Note that, while our model can synthesize the background, we only evaluate the quality of the foreground synthesis for consistent and fair comparison across baseline methods where we use existing segmentation

Table 2: Quantitative results of person-specific animation. Each of the three values are in the order of LPIPS($\downarrow$)$\times 10^2$, tLPIPS($\downarrow$)$\times 10^3$, and SSIM($\uparrow$)$\times 10$, respectively. The number of images used for training is indicated in parentheses, e.g., (6K).

| Methods | Data 1 (6K) | Data 2 (10K) | Data 3 (10K) | Data 4 (15K) | Data 5 (15K) | Average |
|---------|-------------|--------------|--------------|--------------|--------------|---------|
| V2V | 1.84/2.95/9.69 | 3.03/3.83/9.60 | 11.51/3.80/9.05 | 3.06/2.98/9.40 | 4.01/4.04/9.49 | 4.69/3.52/9.45 |
| EDN | 2.74/3.86/9.57 | 3.98/5.40/9.46 | 13.12/4.52/8.96 | 4.90/5.09/9.22 | 5.00/4.82/9.34 | 5.95/4.74/9.31 |
| HFMT | 3.68/4.41/9.48 | 6.39/8.54/9.26 | 13.27/4.62/8.91 | 6.08/3.22/9.10 | 6.86/4.53/9.21 | 7.26/5.06/9.19 |
| DIW | 1.83/2.88/9.68 | 2.70/4.11/9.61 | 11.89/4.09/9.03 | 2.83/4.66/9.45 | 4.14/5.20/9.48 | 4.68/4.19/9.45 |
| MDMT | **1.76/2.58/9.73** | **2.68**/3.77/9.65 | 10.48/3.12/9.11 | 2.81/2.86/9.45 | **3.81/4.12**/9.50 | **4.31**/3.29/9.49 |
| Ours | 1.90/2.83/9.72 | 2.91/**3.76/9.65** | **10.32/2.52/9.28** | **2.78/2.85/9.57** | 4.07/4.26/**9.51** | 4.39/**3.22/9.55** |

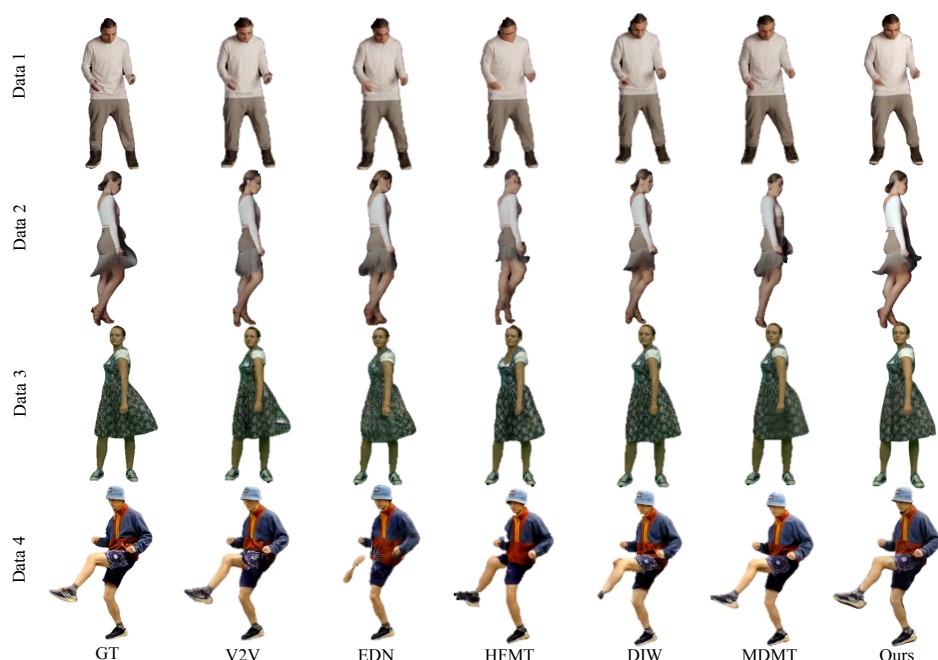

Figure 6: Qualitative comparison between the state-of-the-art methods and our approach for the task of generating person-specific human animation.

method Gong et al. (2018) to remove the background. for evaluation. Specifically, our method outshines all others in the SSIM evaluation with a diffusion-based generative framework. The highest average score implies that our method performs consistently better than other methods in terms of temporal coherence and visual plausibility across assorted appearance and motion styles.

Further qualitative results are demonstrated in Figure 6 where the baseline methods often lose context or become blurred in complex poses, leading to physically implausible human animation. Our method demonstrates robustness to dynamic movements and strong temporal coherence, yielding clear and stable results. Please also refer to the demo video.

### 4.3 ABLATION STUDY

To evaluate the effect of the module in our method, we perform an ablation study. The quantitative results are in Table 3, and please refer to the supplementary materials for visual comparison.

**Unidirectional vs. Bidirectional** We compare our bidirectional temporal diffusion model with the unidirectional one. For this, we trained the same BTU-Net in a unidirectional manner using the same loss. The bidirectional approach demonstrates far more spatiotemporal consistency based on tLPIPS in Table 3. Our main observation is that due to significant motion-appearance ambiguity, the generated texture sometimes diverges at the end of the frame under highly dynamic human

Table 3: Quantitative results on ablation study with graphics simulation data.

| Methods | SSIM↑ | LPIPS↓ | tLPIPS↓ |
|---|---|---|---|
| Ours w/ Unidirectional Model | 0.937 | 0.052 | 0.005 |
| Ours | **0.958** | **0.036** | **0.003** |

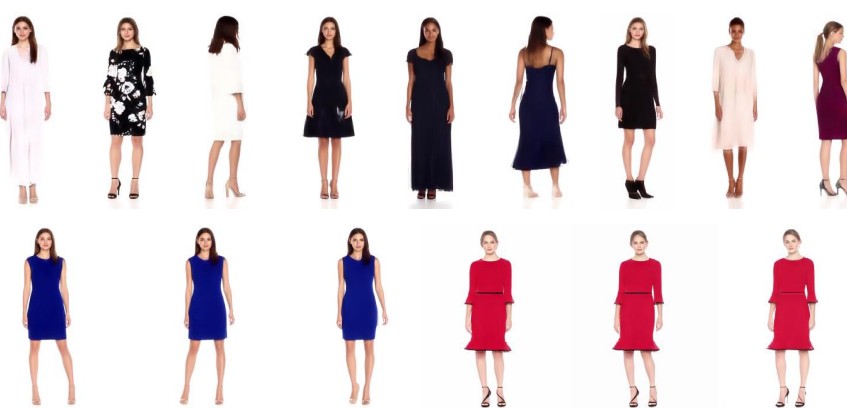

Figure 7: Qualitative results of unconditional animation generation with UBC Fashion data. The top row shows various people generated unconditionally, while the bottom row displays samples of the generated sequences.

movements. Based on the improvements in LPIPS, we can notice such strong temporal coherence helps with improving the visual quality as well.

**Number of Images for Fine-Tuning** For single-image animation, we fine-tune the model on a single image. We found that the quality improves as the number of fine-tuning images increases. In the supplementary material, we introduce the experiments about the impact of the number of fine-tuning images on rendering quality.

**Unconditional Human Animation Generation** Figure 7, we demonstrate that our method can generate human animation with diverse appearances without conditioning any images or videos.

## 5    CONCLUSION

We introduce a new method to synthesize temporally coherent human animation from a single image, a video, or a random noise. We address the core challenge of temporal incoherence from existing generative networks that decode future frames in an auto-regressive way. We argue that such unidirectional temporal modeling of a generative network involves a significant amount of motion-appearance ambiguity, leading to the artifacts such as texture drifting. We suppress the motion-appearance ambiguity by newly designing a bidirectional temporal diffusion model (BTDM): a denoising network progressively removes temporal Gaussian noises whose intermediate results are cross-conditioned over consecutive frames, which allows conditioning locally and globally coherent motion context on our video generation framework. We perform the evaluation on two different tasks, i.e., human animation from a single image and person-specific human animation, and demonstrate that BTDM shows strong temporal coherence, which also helps to improve the visual quality, compared to existing methods.

**Limitation** While BTDM produces temporally coherent human animations, there exist several limitations. Since our model generates the video as a function of the estimated body poses, the errors in the pose estimation affect the rendering quality, *e.g.,* the misdetection of hands produces some appearance distortion around the hand. Due to the inherent ambiguity of 2D pose representation, our method sometimes shows weakness in the sequence with 3D human rotations. Our potential future work is to improve 3D awareness and completeness by utilizing a complete 3D body model, e.g., SMPL Loper et al. (2015), in our bidirectional temporal diffusion framework.

## 6 ACKNOWLEDGEMENT

This work was supported by Institute of Information communications Technology Planning Evaluation (IITP) grant funded by the Korea government(MSIT) (No.RS2023-00225630, Development of Artificial Intelligence for Text-based 3D Movie Generation).

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

## A  BIDIRECTIONAL RECURSIVE SAMPLING

To effectively utilize our Bidirectional Temporal Diffusion Model (BTDM) during the inference stage, we have employed a bidirectional recursive sampling method, which proceeds as follows.

---
**Algorithm 1** Bidirectional Recursive Sampling

---
**Input:** Initial noisy inputs $Y^k = \{y_1^k, ..., y_t^k\}$, driven pose sequence $\mathcal{S} = \{s_1, ..., s_T\}$.
**Output:** Denoised animation $Y^0 = \{y_1^0, ..., y_t^0\}$
**for** $k = K - 1$ **to** $0$ **step** $-1$ **do**
  **if** $K - k$ **is odd then**
    **Direction:** Forward
    **for** $t = 1$ **to** $T$ **do**
      $y_t^{k-1} = f_\theta(y_t^k, y_{t-1}^k, \lambda(k), s_t, d_f)$
    **end for**
  **else**
    **Direction:** Backward
    **for** $t = T$ **to** $1$ **step** $-1$ **do**
      $y_{t-1}^{k-1} = f_\theta(y_{t-1}^k, y_t^k, \lambda(k), s_{t-1}, d_b)$
    **end for**
  **end if**
**end for**

---

Although it's possible to reverse the entire sequence (starting in the backward direction and then moving to the forward, followed by backward again), we observed no significant differences in the outcomes between these two cases.

## B    IMPLEMENTATION DETAILS

### B.1    SINGLE IMAGE ANIMATION

Our method is trained at a resolution of 256x256, similar to all other methods. We generates 64x64 animations via the BTU-Net, which are subsequently upscaled to 256x256 using the SR3 Saharia et al. (2022b).We trained both the BTU-Net and SR3 from scratch on the entire training dataset, for 50k and 100k iterations with a batch size of 32, respectively. We set the denoising step to $K = 1000$ and the learning rate to 1e-5. During testing, we fine-tune model with test appearance condition for 300 iterations with a learning rate of 1e-5. It should be noted that we employ $K = 50$ at test time for expedited generation.

### B.2    PERSON SPECIFIC ANIMATION

The training settings for the BTU-Net and SR3 Saharia et al. (2022b) are identical to those used in the Single image animation setup, with the exception that both the BTU-Net and SR3 Saharia et al. (2022b) are trained for 100 epochs each without fine-tuning.

### B.3    BTU-NET ARCHITECTURE

The detailed structural information of the BTU-Net's layers is illustrated in Figure 8. Due to the complexity of the arrows, the input directions for $E_a$ and $E_p$ have been omitted. Directions are provided in Figure 4 of main manuscript. The notation '$\times digit$' below the dashed block indicates how many times that block structure is repeated.

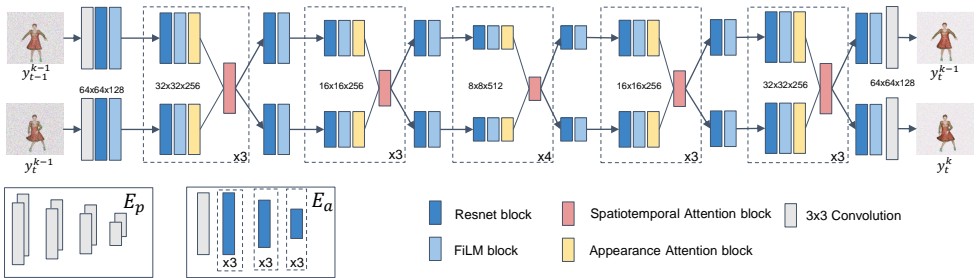

Figure 8: Architecture of BTU-Net.

### B.4    TRAINING AND INFERENCE SPEED ANALYSIS

In particular, training the BTU-Net and SR3 models using the UBC fashion dataset requires 15 and 30 epochs, respectively, on a setup of four A100 GPUs, completed within 67 hours. Fine-tuning these models for 300 iterations takes approximately 210 seconds. During inference, processing each frame takes roughly 1.4 to 1.9 seconds on a single A100 GPU.

## C    ABLATION STUDY

### C.1    COMPARISON OF BIDIRECTIONAL AND UNIDIRECTIONAL APPROACHES

Figure 9 demonstrates the qualitative results of bidirectional and unidirectional temporal training via our BTU-Net. The unidirectional approach struggles to generate images fitting the pose condition, tending to replicate the texture of the front image input as the condition instead. Unlike the unidirectional approach, the bidirectional model successfully creates images that meet the pose condition.

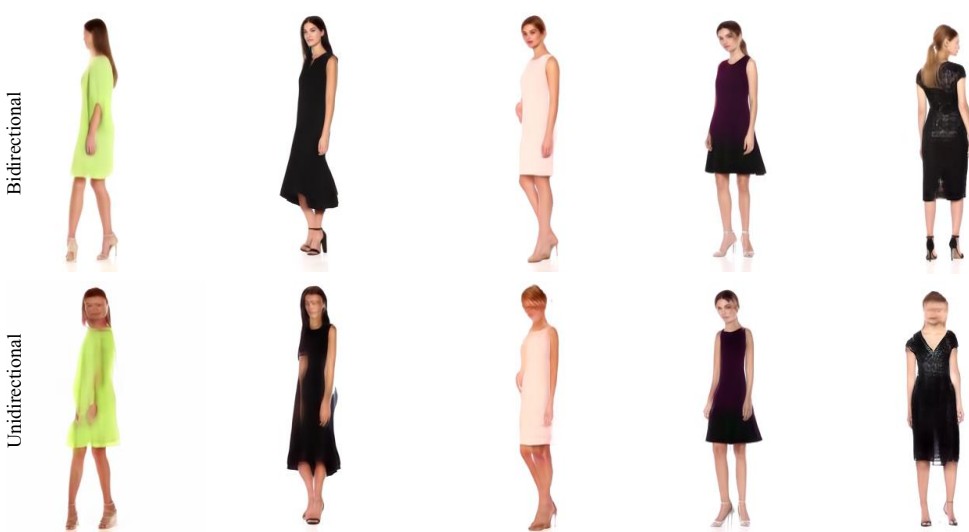

Figure 9: Qualitative comparison of Bidirectional and Unidirectional methods.

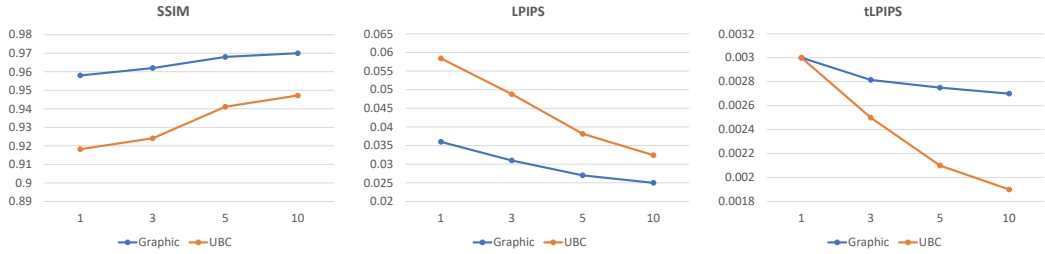

Figure 10: Quantitative comparison of the number of images used for fine-tuning. "Graphic" and "UBC" notates graphic simulated data and UBC fashion data results respectively.

## C.2 NUMBER OF IMAGES FOR FINE-TUNING

We also evaluate the performance depending on the number of images used for fine-tuning. The performance comparison results are shown in Figure 10 and Figure 11. As the number of images used for fine-tuning increases, performance improves across various metrics. Notably, the trend between the increase in image count and metric scores is not linear but shows signs of convergence.

## C.3 USER STUDY

We conduct quantitative results of an user study in which people evaluated videos generated by our method and baseline methods. A total of 42 participants took part in this study, which involved tasks for *single image animation* and *person specific animation*. Each evaluation required participants to watch comparison videos at least twice and make selections based on two questions: "*Which video preserves the identity best?*" and "*Which video looks most realistic to you?*". For the person specific animation task, the experiment was conducted excluding the first question. As shown in Figure 12, it can be seen that our BTMD results are much more realistic and maintain identity better when evaluated by people, compared to other baseline methods.

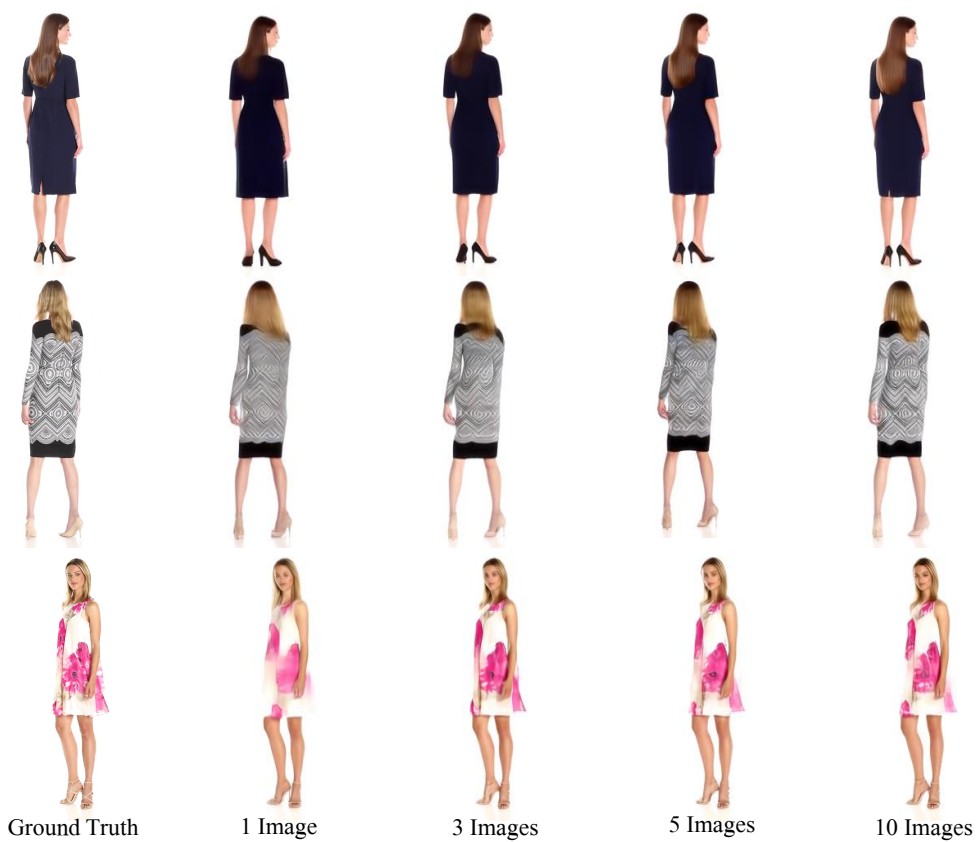

Figure 11: Qualitative comparison of the number of images used for fine-tuning.

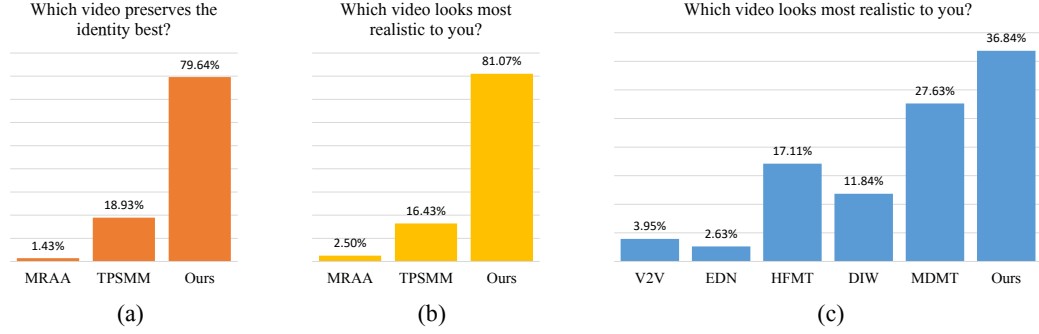

Figure 12: Quantitative result of human evaluations. Graphs (a) and (b) represent the results for the *single image animation* task, showing the proportion of choices made for two different questions. Graph (c) shows the results for the *person-specific animation* task.

## C.4 UNCONDITIONAL ANIMATION GENERATION

We demonstrate that our method can generate human animations featuring diverse clothing styles and identities, even without any image conditions. Results from unconditional generation experiments on both datasets are illustrated in Figures 13 and 14.

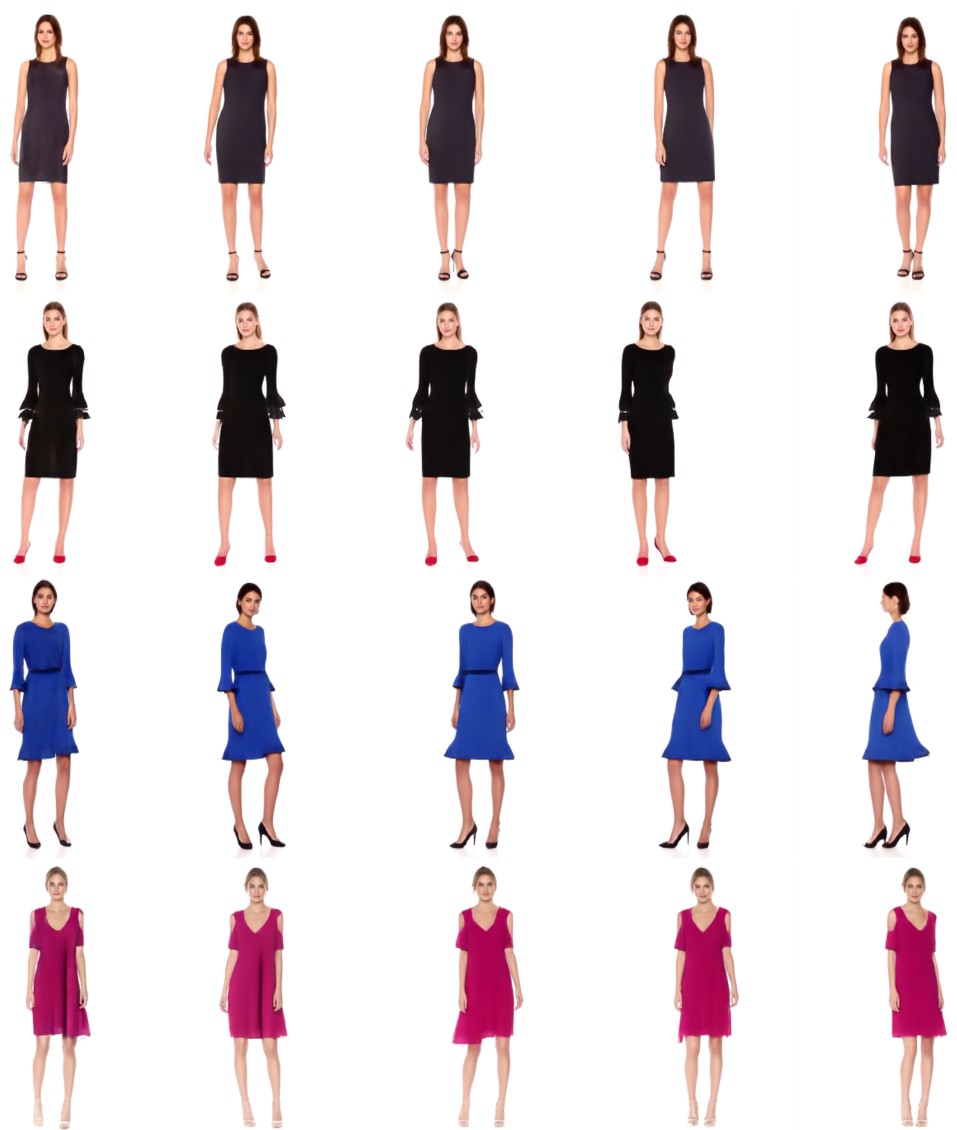

Figure 13: Sample sequences from unconditional generation on UBC fashion data.

# D  GRAPHIC SIMULATED DATASET

Graphic simulated dataset is comprised of approximately 98,000 images, each rendered at a resolution of 512x512. These images illustrate various dynamic movements (such as dance, exercise, etc.) of 3D human models with a total of 99 different appearances. The 3D human models in this dataset are created using Character Creator 4 Reallusion (a), and we simulate the soft cloth motion in iClone8 Reallusion (b). Mixamo Adobe human motions, are exported as an Alembic Pictures file. For realistic rendering, we employ Ray Tracing Texel (RTX) rendering and the Nvidia Omniverse Nvidia as the rendering tool. Figure 15 shows a few samples from our graphically simulated data.

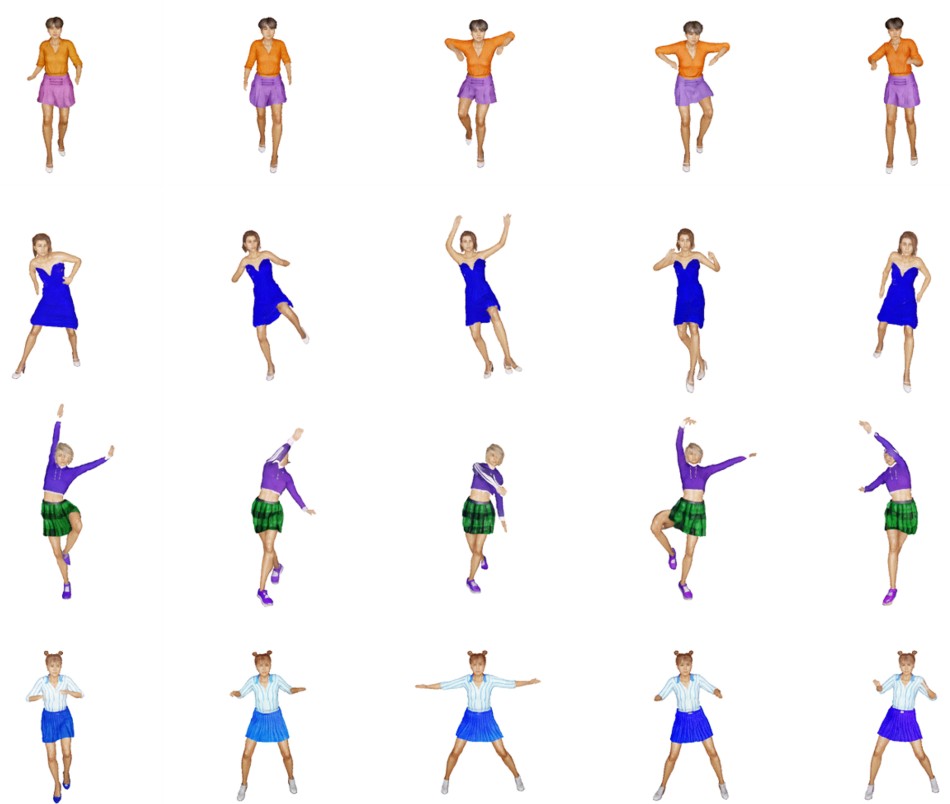

Figure 14: Samples from unconditional generation on Graphic simulated data.

# E   MORE VISUAL RESULTS

## E.1   SINGLE IMAGE ANIMATION

Additional visual results conducted with graphic simulated and UBC fashion data for the single image animation task are shown in Figures 17 and 18.

## E.2   PERSON-SPECIFIC ANIMATION

For a fair comparison evaluation in the person-specific human animation task, we evaluate the results using only the foreground. However, our method is capable of background synthesis, and the visual results are shown in Figure 16.

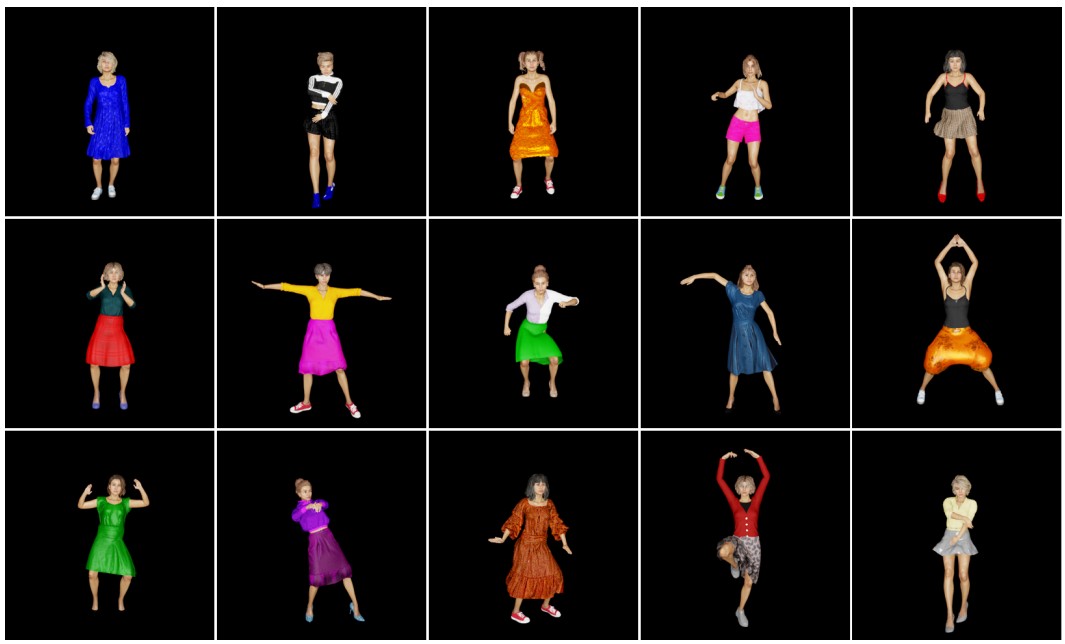

Figure 15: Samples of Graphic Simulation Dataset

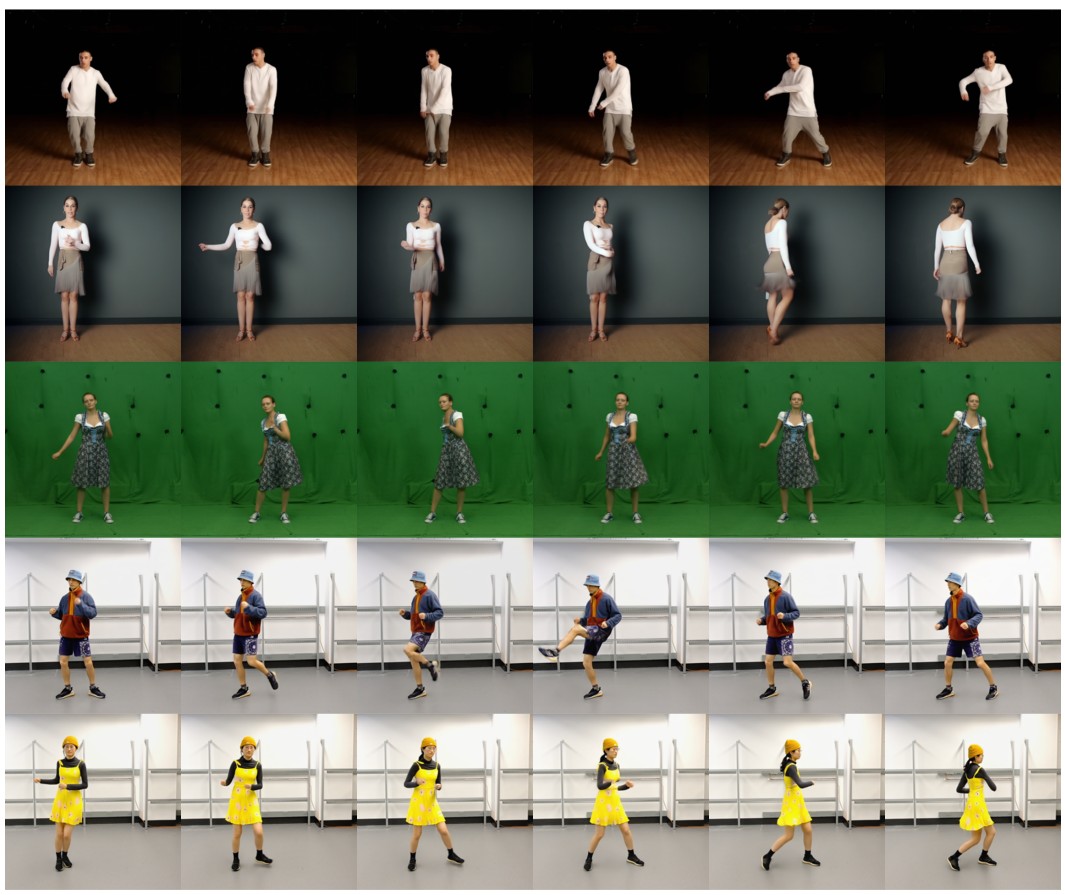

Figure 16: Samples from person-specific animation results with background.

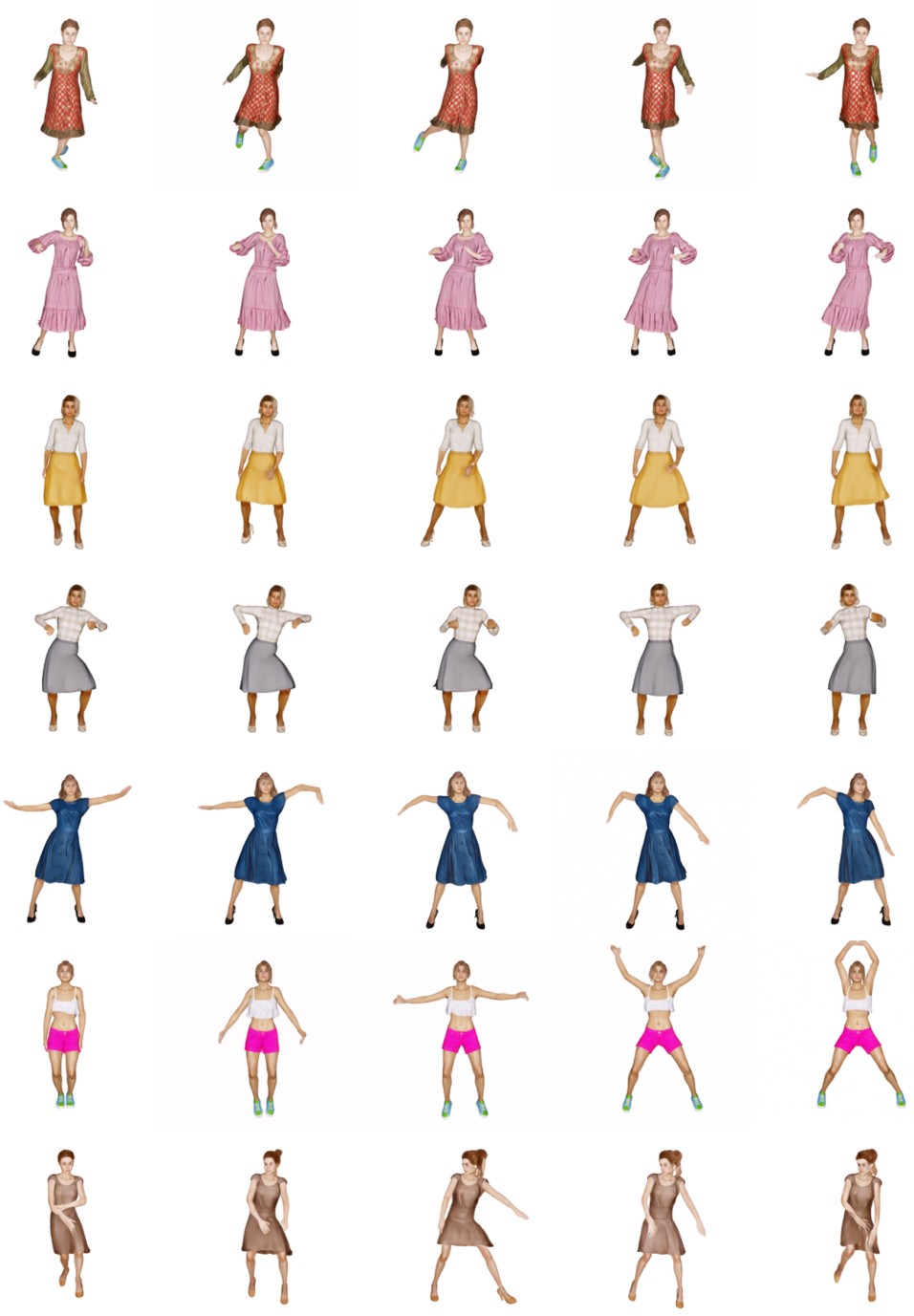

Figure 17: More single image animation results on Graphic simulated data.

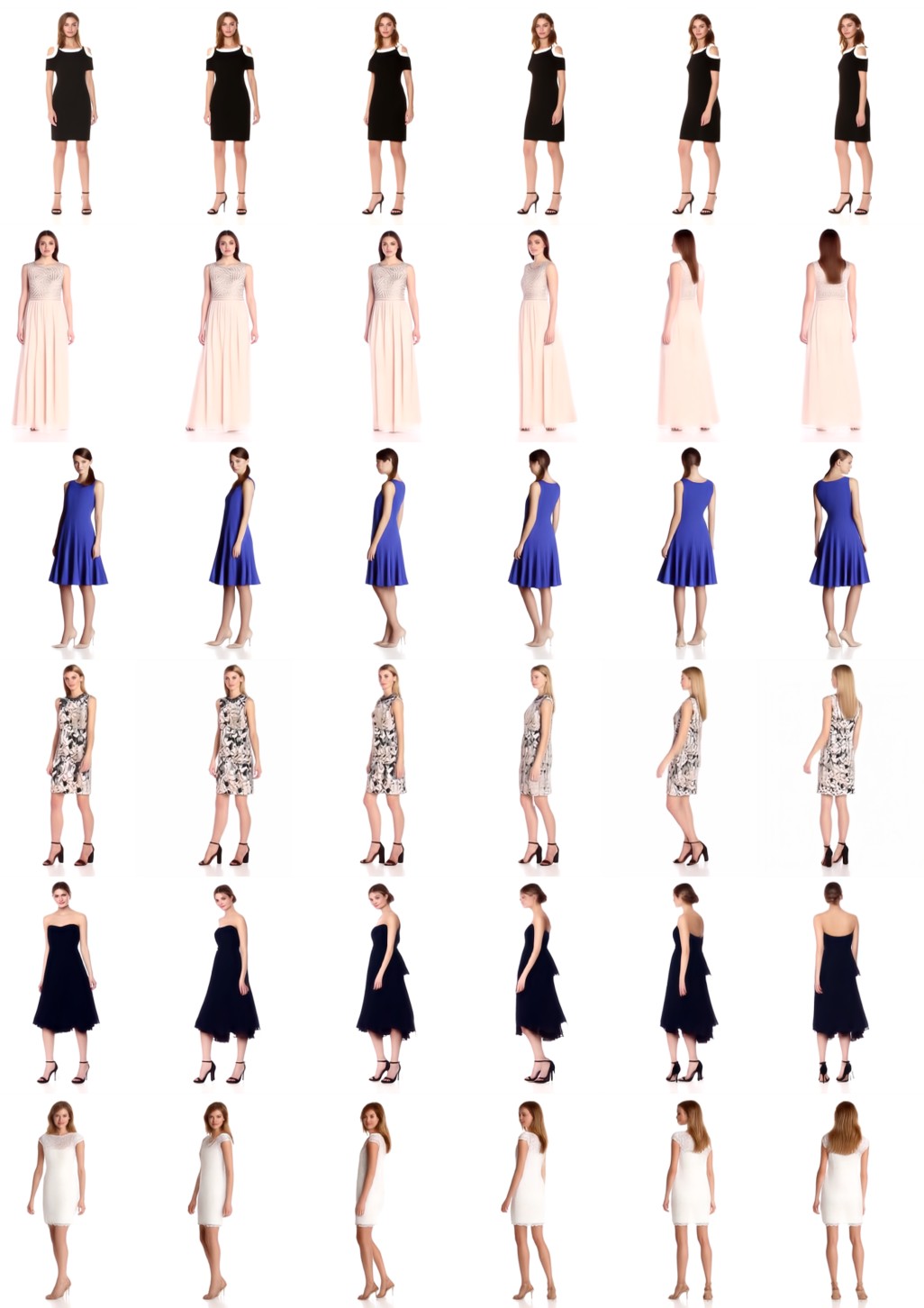

Figure 18: More single image animation results on UBC fashion data.

