# OpenReview forum: "Bidirectional Temporal Diffusion Model for Temporally Consistent Human Animation"
_ICLR.cc/2024/Conference — ICLR 2024 poster_

### Official Review · Reviewer_xCTh · 2023-10-30

**Soundness:** 3 good
**Presentation:** 3 good
**Contribution:** 3 good
**Rating:** 6
**Confidence:** 4

**Summary:**

In this paper, the authors design a new human animation framework using a denoising diffusion probabilistic model, leveraging bidirectional temporal modeling and a denoising diffusion model to enhance temporal coherence and reduce artifacts in the generated animations. They proposed a bidirectional temporal diffusion model that can generate temporally coherent human animation from random noise, a single image, or a single video. They introduced a feature cross-conditioning mechanism between consecutive frames using recursive sampling to enable local Embed action background information in a globally consistent manner.

**Strengths:**

1. The structure of the article is reasonable, clear, and easy to understand. The design of pictures and tables is rigorous.
2. The method section and experiment settings are introduced in the details.

**Weaknesses:**

- Most of the references are before 2022, and there are no latest research results in 2023.
- Too many arXiv papers are cited in the references. The publication information of the article should be indicated.
- The paper mentions three tasks of generating human animation from a single noise, a single image, and a single video, but the quantitative results only include the single image animation.
- More quantitative results and visual examples should be added.

**Questions:**

Please see the questions above.

---

> ### Author Response · Authors · 2023-11-20
>
> Thank you for providing valuable feedback that can further enhance our paper. Based on your comments, we will provide a detailed response.
>
> **Q1**: Most of the references are before 2022, and there are no latest research results in 2023.
>
> **A1**: Thank you for your suggestion.  In the revision, we added the discussion about more recent works in the related work section.
>
> **Q2**: Too many arXiv papers are cited in the references. The publication information of the article should be indicated.
>
> **A2**: Thank you for pointing out the aspect we missed. As you suggested, we have edited the publication information in the revision.
>
> **Q3**: The paper mentions three tasks of generating human animation from a single noise, a single image, and a single video, but the quantitative results only include the single image animation.
>
> **A3**: We would like to politely inform the reviewer that the quantitative results for the person-specific animation (a single video) can be found in Table 2.
>  As for the noise-to-animation case, we have not conducted a quantitative analysis  since generating a human animation from a noise is one of our “application”, not the main idea in the paper. Also there is no appropriate baseline to compare for generating human animations from noise.
>
> **Q4**: More quantitative results and visual examples should be added.
>
> **A4**:  For more quantitative results, we perform a user study as described in the appendix (C.3) of the revision. A total of 42 participants took part in this experiment, which involved tasks for the single image animation and person specific animation. Each experiment required participants to watch comparison videos at least twice and make selections based on two questions: “Which video preserves the identity best?” and “Which video looks most realistic to you?”. For the person specific animation task, the experiment was conducted excluding the first question. The analysis of the experiment results has been added to appendix (C.3).
>
> Following your suggestion, We have also added visual examples to the appendix (E) of the revision.

---

### Official Review · Reviewer_rYhA · 2023-10-31

**Soundness:** 2 fair
**Presentation:** 3 good
**Contribution:** 3 good
**Rating:** 6
**Confidence:** 4

**Summary:**

This paper aims to generate temporally coherent human animation from a single image, a video or a random noise. The authors propose the bidirectional temporal modeling to improve the temporal consistency, which is different from existing unidirectional formulation. Specifically, a diffusion based generative network is designed to decode the human appearance in both forward and backward direction, where  intermediate features are cross-conditioned over time. Experimental results on different generation tasks show better performance of the proposed method compared with existing unidirectional based methods.

**Strengths:**

1. The paper proposes a novel human animation framework, which models the human appearance bidirectionally to improve the appearance consistency within a video sequence.
2. The proposed method achieves better performance and generalization capacity compared with existing works.

**Weaknesses:**

1. The results in Table 2 are not consistent with the reported results in MDMT. It would be better to explain the reason.
2. The main objective of the paper is to improve the temporal appearance coherence of the generated video sequence. The reviewer is wondering how is the proposed bidirectional modeling compared to the texture inversion [A, B] technique in keeping the appearance consistent.
[A] Nataniel Ruiz et al. DreamBooth: Fine Tuning Text-to-Image Diffusion Models for Subject-Driven Generation.
[B] Rinon Gal et al. An Image is Worth One Word: Personalizing Text-to-Image Generation using Textual Inversion.
3. Related works on diffusion model based video generation are missing, such as NUWA, Make-A-Video, MagicVideo and so on, especially discussions on how these works enforce the temporal consistency.
4. The intuition on how the proposed bidirectional temporal modeling solves the motion-appearance ambiguity problem needs to be further explained.
5. Typo in the line above Eq.(4), should be p(y_(t-1)/y_t) p(y_(t-1)/p_t)?

**Questions:**

Please refer to the weaknesses.

---

> ### Author Response · Authors · 2023-11-20
>
> Thank you for providing valuable feedback that can further enhance our paper. Based on your comments, we will provide a detailed response.
>
> **Q1**: The results in Table 2 are not consistent with the reported results in MDMT. It would be better to explain the reason.
>
> **A1**: Due to limitations with our GPU resources, we conducted our evaluations using a resolution of 256x256, in contrast to the 512x512 resolution used in MDMT, resulting in different numerical outcomes. However, aside from this difference in resolution, all other aspects of our evaluation followed the same protocol as MDMT. We would like to note that all the data used for our comparative analysis were provided by the authors of MDMT.
>
> **Q2**: The reviewer is wondering how is the proposed bidirectional modeling compared to the texture inversion [A, B] technique in keeping the appearance consistent.
>
> [A] Nataniel Ruiz et al. DreamBooth: Fine Tuning Text-to-Image Diffusion Models for Subject-Driven Generation.
>
> [B] Rinon Gal et al. An Image is Worth One Word: Personalizing Text-to-Image Generation using Textual Inversion.
>
> **A2**: The papers [A, B] referenced by the reviewer indeed offer valuable insights into appearance consistency. However, due to the fundamental task difference between our work (i.e., pose-guided video generation focusing on temporal consistency) and [A,B] (i.e., text-to-image generation), we respectfully point out that it is not possible to directly compare our BTDM model with the one from [A,B].
>
> **Q3**: Related works on diffusion model based video generation are missing, such as NUWA, Make-A-Video, MagicVideo and so on, especially discussions on how these works enforce the temporal consistency.
>
> **A3**: Thank you for your suggestion. We added the suggested works in the related works with some discussion.
>
> **Q4**: The intuition on how the proposed bidirectional temporal modeling solves the motion-appearance ambiguity problem needs to be further explained.
>
> **A4**: The conventional unidirectional auto-regressive approach uses the result of the previous time step as a condition to generate the frame of the next time step. However, if an erroneous result (e.g., appearance distortion and blur) is produced at a previous time step, it drift to the next time step due to the motion-appearance ambiguity, leading to the results whose errors are magnified as shown in Figure 2 of the main paper, Figure 9 of the appendix, and in our supplementary videos. In contrast, our bidirectional temporal method is designed to highly suppress such motion-appearance ambiguity by performing temporally bidirectional cross-conditioning between adjacent frames in the denoising diffusion framework. This method encodes correlations between adjacent frames and throughout the sequence via Markovian independence. By propagating information bidirectionally and temporally, we enhance both local and global consistency. This ensures the generation of the temporally coherent and drift-free human animation.
>
> **Q5**: Typo in the line above Eq.(4), should be p(y_(t-1)/y_t) p(y_(t-1)/p_t)?
>
> **A5**: Thank you for pointing out the aspect we missed. We have made the necessary corrections to address this.

---

### Official Review · Reviewer_mGPz · 2023-11-01

**Soundness:** 3 good
**Presentation:** 3 good
**Contribution:** 3 good
**Rating:** 8
**Confidence:** 4

**Summary:**

This paper proposes a novel method for human animation from a single image, a video or a random noise.
- The main contribution is bidirectional temporal modeling which can generate temporally coherent videos.
- Both qualitative and quantitative results show the effectiveness of proposed method.

**Strengths:**

+ The paper is well written and easy to understand.
+ The bidirectional temporal modeling is novel and effective in improving the quality of generated videos.
+ The proposed bidirectional diffusion process to overcome overfitting and the bidirectional attention block are both reasonable.
+ The experiments are sufficient and demonstrate the effectiveness of the proposed method.

**Weaknesses:**

- The datasets used for experiments are simple because the background is white. It is not clear how the proposed method works on real world scenario where the background is complicated.
- Table 1, tOF does not follow the format of other metrics.

**Questions:**

- How does the paper compare with first order motion model (Neurips 2019) which has been a strong baseline in image animation?
- How long does it take to generate a video in the experiments? Is it time consuming using a bidirectional diffusion model?

---

> ### Author Response · Authors · 2023-11-20
>
> Thank you for providing valuable feedback that can further enhance our paper. Based on your comments, we will provide a detailed response.
>
> **Q1**: The datasets used for experiments are simple because the background is white. It is not clear how the proposed method works on real world scenario where the background is complicated.
>
> **A1**: This work is designed to mitigate motion-appearance ambiguity to enhance temporal consistency of the foreground. Therefore, it is important to note that all experiments were conducted without considering the background elements (i.e., by removing the background using an existing foreground detection method as a pre-processing).
> Still, however, to address the reviewer’s comment, we trained our BTDM with background-inclusive data for person-specific animation task, and have added visual results to the appendix (E) where our method is able to jointly learn the foreground and background.
>
> **Q2**: Table 1, tOF does not follow the format of other metrics.
>
> **A2**: Thank you for pointing out the aspect we missed. We have made the necessary corrections to address this in the revision.
>
> **Q3**: How does the paper compare with first order motion model (Neurips 2019) which has been a strong baseline in image animation?
>
> **A3**: Thank you for your suggestion. Indeed, the First Order Motion Model (FOMM) could be a viable candidate for the comparison. However, we did not include it as a baseline because both the Motion Representations for Articulated Animation (MRAA) [Siarohin et al. 2021] and the Thin-Plate Spline Motion Model for Image Animation (TPSMM) [Zhao et al. 2022] have already demonstrated superior performance in various metrics compared to FOMM.
>
> **Q4**: How long does it take to generate a video in the experiments? Is it time consuming using a bidirectional diffusion model?
>
> **A4**: During inference, processing takes roughly 1.4 to 1.9 seconds per frame on a single A100 GPU, which is proportional to the number of frames. The speed analysis has been added to the appendix (B.4). We kindly note that our method, due to performing inference on adjacent frames in parallel, may have slower inference speed compared to conventional image generation diffusion models.

---

### Official Review · Reviewer_VxE3 · 2023-11-06

**Soundness:** 2 fair
**Presentation:** 3 good
**Contribution:** 2 fair
**Rating:** 6
**Confidence:** 3

**Summary:**

The paper introduces a method that generates animated human videos. Depending on the training protocol, the proposed method can generation human animations either unconditionally, or conditionally from images (single image animation) or videos (Person-specific animation). Compared to previous works that relies on autoregressive generation of frames, this paper proposes a bidirectional diffusion model, achieving better quality. The bidirection model takes two noisy frames as input, denoising one of them according to the input embedding that controls the direction.
To facilitate the evaluation of the model, a synthetic human animation dataset containing different characters having the same motion is built.

**Strengths:**

* The proposed method outperforms prior art on a significant number of benchmarks.
* The paper also designed a new, high quality synthetic dataset for better evaluation of the model. This dataset, if released, can be of great help to future works.

**Weaknesses:**

* The proposed bidirectional model is not new. It closely resembles a video diffusion model with temporal cross attention -- a standard technique used in video generation.
* The writting of the paper could be improved. For example, it is unclear how the two directions of the bidirectional model is combined during inferenced. This information is very important for the understanding of the model.

**Questions:**

* The BTU-Net described in the paper has a temporal window of 2. Would increasing the number of frames further improve the performance?
* How is the proposed model inferenced? Do you take the average of the results from both directions?

---

> ### Author Response · Authors · 2023-11-20
>
> Thank you for providing valuable feedback that can further enhance our paper. Based on your comments, we will provide a detailed response.
>
> **Q1**: The proposed bidirectional model is not new. It closely resembles a video diffusion model with temporal cross attention a standard technique used in video generation.
>
> **A1**: We agree that our method is not the first approach in the diffusion-based video generation framework. However, unlike existing “video generation” works [A,B,C,D,E], we propose a novel approach that can embed the bidirectional context over all frames using Markovian independence property in the frame-by-frame “image generation” framework.
>
> The current video diffusion models [A,B,C,D,E] are based on 3D (pseudo 3D) convolution and attention which are highly limited in decoding a long video sequence due to inherent spatial (i.e., resolution) and temporal (i.e., number of frames) resource constraints. In our case, we introduce a novel bidirectional temporal approach in the image generation framework: we uniquely encodes the correlations between adjacent frames (local) and across all frames (global) through bidirectional Markovian independence, which enforce global temporal consistency in the generated human animation regardless of the generated frame length.
>
> [A] Uriel Singer et al. Make-A-Video: Text-to-Video Generation without Text-Video Data.
>
> [B] Daquan Zhou et al. MagicVideo: Efficient Video Generation with Latent Diffusion Models.
>
> [C] Ruihan Yang et al. Diffusion Probabilistic Modeling for Video Generation.
>
> [D] Jonathan Ho et al. Video Diffusion Models.
>
> [E] Xin Li et al. VideoGen: A Reference-Guided Latent Diffusion Approach for High Definition Text-to-Video Generation
>
>
> **Q2**: How is the proposed model inferenced? Do you take the average of the results from both directions?
>
> **A2**: We take the advantage of the results from the forward and backward path by alternating the direction at each denoising step. For example, we start at the denoising stage $k=K-1$, progressing from time $t=1$ to $t=T$ in the forward direction to obtain inference results. Then, we proceed to the $k=K-2$ denoising stage, using the results from $k=K-1$ as inputs for this step. At stage $k=K-2$, inference is conducted in the backward direction, from $t=T$ to $t=1$, and this alternating directional approach continues until reaching $k=0$. Although it's possible to reverse the entire sequence direction (starting in the backward direction and then moving to the forward, followed by backward again), we observed no significant differences in the outcomes between these two cases.
>
> As you mentioned, averaging the results from the forward and backward processes is another way of bidirectional denoising. However, in our experiment,  it led to the observation of a 'ghosting' effect (see supplementary material ghost.png). This approach also has the disadvantage of doubling the inference time. We have included an explanation of inference methodology in the Appendix (A) for further clarity.
>
>
> **Q3**: The BTU-Net described in the paper has a temporal window of 2. Would increasing the number of frames further improve the performance?
>
> **A3**: Increasing a temporal window would not significantly improve the performance since the video generated by our bidirectional diffusion model already captures the global motion context over all frames through the bidirectional Markovian independence chained by adjacent frames.
>
> Also, we kindly note that expanding the window size in our framework is not possible without substantial modification since the structure of our BTDM's frame-by-frame image generation framework should be largely remodeled in a way that incorporates 3D convolution and attention techniques. Such an approach is also different from our intent to use the image generation framework, which is one of our novelties, for generating human animation.

---

> > ### Comment · Reviewer_VxE3 · 2023-11-22
> >
> > Thanks for the clarification as well as the updated paper! The method makes sense now, especially when the computation and memory is limited and cross-attention methods become too expensive. I have thus raised my score.

---

> ### Author Response · Authors · 2023-11-23
>
> We're glad to hear that the updated paper and our explanation about the method were helpful to the reviewer. We appreciate your willingness to reconsider and raise the score. If you have any more questions or need further clarification, please feel free to ask.

---

### Meta-Review · Area_Chair_42Ke · 2023-12-12

**Metareview:**

The paper introducing an algorithm for generating human motion that is consistent across time by proposing bi-directional constraints, thus forcing the model to be temporally cohesive from past to future and from future to past. Further, under a Markovian assumption the model is forced to be 'globally' coherent. The problem with autoregressive approaches is that errors accumulate over time and there is no constraint making sure that errors are suppressed. By contrast, bi-directionality enforces a smoothness constraint. A new dataset is also introduce on human motion. All reviewers agree that the paper is of value, with good results, and reasonable novelty, thus I recommend acceptance.

**Justification For Why Not Higher Score:**

Novelty is ok bu limited.

**Justification For Why Not Lower Score:**

Good results, reasonable novelty.

---

### Decision · Program_Chairs · 2024-01-16

Accept (poster)